# Could Bone Biomarkers Predict Bone Turnover after Kidney Transplantation?—A Proof-of-Concept Study

**DOI:** 10.3390/jcm11020457

**Published:** 2022-01-17

**Authors:** Juliana Magalhães, Janete Quelhas-Santos, Luciano Pereira, Ricardo Neto, Inês Castro-Ferreira, Sandra Martins, João Miguel Frazão, Catarina Carvalho

**Affiliations:** 1Nephrology and Infectious Diseases Research Group, Institute for Innovation and Health Research (I3S), Institute of Biomedical Engineering (INEB), University of Porto, 4200-135 Porto, Portugal; jucmagalhaes@gmail.com (J.M.); lucianoarturpereira@hotmail.com (L.P.); ricardoneto76@gmail.com (R.N.); inescastroferreira@sapo.pt (I.C.-F.); jmmdfrazao@gmail.com (J.M.F.); 2Faculty of Medicine, University of Porto, 4200-250 Porto, Portugal; sjanete@med.up.pt; 3Nephrology Department, Faculty of Medicine, University of Porto, 4200-250 Porto, Portugal; 4Centro Hospitalar de São João and EPI Unit, Clinical Pathology Department, Institute of Public Health, University of Porto, 4200-319 Porto, Portugal; smvrmartins@gmail.com

**Keywords:** bone biomarkers, kidney transplantation, bone turnover

## Abstract

Aim: Bone disease after kidney transplant (KT) results from multiple factors, including previous bone and mineral metabolism disturbances and effects of transplant-related medications. New biomolecules have been recently associated with the development and progression of the chronic kidney disease–associated bone and mineral disorder (CKD-MBD). These include sclerostin and the soluble receptor activator of nuclear factor-kB ligand (sRANKL). Methods: To better understand the role of biomarkers in post-transplant bone disease, this study was designed to prospectively evaluate and correlate results from the histomorphometric analysis of bone biopsies after KT with emerging serum biomarkers of the CKD-MBD: sclerostin, Dickkopf-related protein 1 (Dkk-1), sRANKL and osteo-protegerin (OPG). Results: Our data shows a significant increase in plasma levels of bioactive sclerostin after KT accompanied by a significant reduction in plasma levels of Dkk-1, suggesting a promotion of the inhibition of bone formation by osteoblasts through the activation of these inhibitors of the Wnt signaling pathway. In addition, we found a significant increase in plasma levels of free sRANKL after KT accompanied by a significant reduction in plasma levels of its decoy receptor OPG, suggesting an enhanced bone resorption by osteoclasts mediated by this mechanism. Conclusions: Taken together, these results suggest that the loss of bone volume observed after KT could be explain mainly by the inhibition of bone formation mediated by sclerostin accompanied by an enhanced bone resorption mediated by sRANKL.

## 1. Introduction

After kidney transplantation (KT), disorders of bone metabolism are important causes of morbidity and mortality [1]. Post-transplant bone disease is characterized by changes in bone quality and density, but also in mineral metabolism, and appears to have multifactorial etiology. Pre-existing renal osteodystrophy at the time of KT seems to be a main factor, but the use of immunosuppressive medications, development of hypophosphatemia and hypercalcemia, and persistent hyperparathyroidism and vitamin D deficiency, also seem to contribute [1,2,3,4]. The risk of fracture in KT population is about 4.8 times higher when compared to the general population (almost 23 times for vertebral fractures [5]) and is estimated to be 30% higher in the first 3 years after transplantation, with a reduction afterwards compared to patients who remain on dialysis [3].

In a previous study by our group, including seven KT patients, the follow-up bone biopsy (repeated after 2–5 years) revealed a significant decrease in osteoblast surface/bone surface, osteoblasts number/bone surface and erosion surface/bone surface, as well as a decrease in trabecular number and increase in trabecular separation. These results show a reduction in bone activity, suggesting a loss in bone quantity [3].

Despite trans-iliac bone biopsy being considered the gold standard when assessing bone metabolism in chronic kidney disease (CKD) patients, new biomolecules have been recently associated with the development and progression of the chronic kidney disease–associated bone and mineral disorder (CKD-MBD). These include sclerostin, a promotor of the inhibition of bone formation by osteoblasts, and the soluble receptor activator of nuclear factor-kB ligand (sRANKL), an enhancer of bone resorption by osteoclast [6,7,8].

### 1.1. Wnt Signaling Pathway: Sclerostin and Dkk-1

Sclerostin is a 22.5 kDa secreted glycoprotein nearly exclusively produced in osteocytes [9] that functions as a potent inhibitor of Wnt signaling. It acts by binding to the Wnt-coreceptor LRP5/6 thus inhibiting bone formation by regulating osteoblast function and promoting osteoblast apoptosis [10,11]. Sclerostin levels are altered in response to hormonal stimuli or due to pathophysiological conditions. New clinical trials in osteoporosis, evaluating sclerostin inhibitors, have shown a self-regulating increase in bone formation and bone mass, at the same time as a prolonged decrease in bone resorption, as well as a decrease in the fracture rate [12]. In CKD patients, sclerostin serum levels are increased up to four-fold compared to patients without CKD, with a progressive increase with declining kidney function [13,14]. Moreover, in dialysis patients, sclerostin seems to be an independent predictor of bone loss [15].

Dickkopf-related protein 1 (Dkk-1) is a member of the Dkk family and is central to embryonic and adult bone development and bone health [16,17]. It is widely expressed in many areas including osteoblasts and osteocytes, as well as the skin and endothelium [17] and is well characterized as an antagonist of canonical Wnt signaling via binding to Wnt-coreceptor LRP5/6. It is known that loss of function mutations in the Wnt co-receptor, LRP5 can lead to osteoporosis and that gain-of-function mutations can lead to high bone mass [18]. It has also been shown that a reduction in Dkk-1 expression can lead to an increase in trabecular and cortical bone mass in vivo [19].

### 1.2. Rank/Rank L/Opg System

sRANKL, a member of the tumor necrosis factor (TNF) family, is the main stimulatory factor for the formation of mature osteoclasts and is essential for their survival. RANKL activates its specific receptor RANK, located on osteoclasts and dendritic cells, promoting bone resorption [20]. The effects are counteracted by OPG, which acts as an endogenous soluble receptor antagonist, therefore decreasing osteoclast resorptive activity. Studies of RANKL and OPG levels in patients with CKD and hemodialysis have shown contradictory results, with increased OPG levels and RANKL levels that can range from high to low when compared to healthy controls, requiring further studies [21,22].

The correlation between these biomarkers and bone histomorphometry in KT recipients is still poorly understood.

This study aimed to prospectively evaluate and correlate the results of histomorphometric analysis of bone biopsies after kidney transplantation, observed and discussed in our first paper, with emerging serum biomarkers of the CKD-MBD spectrum: Sclerostin, Dkk-1, sRANKL and OPG. Our aim also was to compare the longitudinal evolution of these new biomolecules with established biomarkers.

## 2. Methods

### 2.1. Patients

Thirteen consecutive patients submitted to deceased donor KT at our institution between 2008 and 2010, who accepted participation in the protocol and presented excellent graft function at time of discharge, were included. Patients previously on dialysis for less than 12 months, who had undergone previous KT or parathyroidectomy, and those treated with bisphosphonates or anticonvulsants were excluded from the study.

All patients received a triple-drug immunosuppressive regimen, consisting of steroid, calcineurin inhibitor and mycophenolate mofetil. Cyclosporine was adjusted to blood levels of 124.8–208 nmol/L in the first 6 months and 66.6–124.8 nmol/L after that period, while tacrolimus was adjusted to values of 8.32–12.48 nmol/L in the initial post-transplant period and 5.82–8.32 nmol/L after 6 months. The prednisone dose was 3400 mg during the first 6 months, changing to an average prednisone dose between 5–10 mg/day after.

All patients presented a well-functioning graft and renal function remained stable during the study, without documented episodes of acute dysfunction.

Six patients agreed to follow-up (FW) evaluation with second biopsy. None of the six patients was treated with calcium supplementation active or native vitamin D, during follow-up. Of these, due to severe hypercalcemia (3 mmol/L), one of the patients started cinacalcet (30 mg/day) at month 20. One patient took bisphosphonates for a short period of 6 months because of low mineral density on ray absorptiometry -Double X (DXA).

The second bone biopsy was performed in 2010–2013. All samples of plasma were immediately stored at −80 °C after collection.

All patients gave their informed consent, which had been previously approved by the Ethics Committee (CE 53-2008).

### 2.2. Bone Biomarkers

Blood samples were collected immediately before KT and at 12, 18 and 24 months after KT and free soluble uncomplexed human RANKL and bioactive sclerostin were measured in plasma samples by an Enzyme Immunoassay according to the manufacturer’s protocol (Biomedica Medi zinprodukte GmbH, standard range 0 to 2 pmol/L and 0 to 320 pmol/L, respectively). Osteo-protegerin/TNFRSF11B (OPG), Dkk-1 and intact FGF-23 were measured in the same plasma samples by a Magnetic Luminex Assay according to manufacturer’s protocol (R&D Systems, Inc. MN, USA, standard range 75 to 18,220 pg/mL, 202 to 49,060 pg/mL and 11.8 to 2870 pg/mL, respectively).

### 2.3. Biochemical Follow-Up

Calcium and phosphorus were measured with standard center assays. Estimated glomerular filtration rate (eGFR) was calculated from serum creatinine levels using the CKD-EPI (Chronic Kidney Disease Epidemiology Collaboration) equation formula [23]. iPTH was measured with an electro-chemiluminescent assay using COBAS e411-Hitachi immunoassay analyzer (Roche Diagnostics GmbH, Mannheim, Germany).

### 2.4. Bone Histomorphometry

Baseline BB was performed at the first 2 months after engraftment in all 13 patients. FW biopsies were performed 24 months after the first biopsy in six patients. Full thickness BB were obtained from the anterior iliac crest using a modified Bordier trephine after double tetracycline labelling as previously described [24]. Biopsy specimens were 0.5 cm in diameter by 1–1.5 cm in length. Specimens were dehydrated in alcohol, cleared with xylene, and embedded in methyl-methacrylate. Serial undecalcified 5 μm sections were obtained and stained with modified Masson-Goldner trichrome for static histomorphometric parameters evaluation. Bone histomorphometric parameters were assessed under 200× magnification using the OsteoMetrics^TM^ system (OsteoMetrics, Decatur, IL, USA). External and internal cortices were analyzed separately. Measured trabecular parameters comply with the nomenclature of the Histomorphometry Nomenclature Committee of the American Society of Bone and Mineral Research [25].

### 2.5. Statistical Analysis

Determinations of bone parameters and biochemical variables are reported as mean + standard error. Paired t-tests were used to evaluate differences in parameters before and after transplant. Spearman correlations were used in correlation analysis. Statistics were computed using excel and GraphPad Prisma. *p* values less than 0.05 were considered statistically significant.

## 3. Results

Demographic characteristics of the patients (summarized on Table 1): mean age was 52.2 years (range 33–64 years), and mean time on renal replacement therapy was 54.9 months (range 24–94 months).

### 3.1. Evolution of Serum Bone-Related Biomarkers after Kidney Transplantation

Figure 1A shows a slight decrease in circulating levels of bioactive sclerostin 12 months after KT compared with baseline levels (1502 ± 201.4 pg/mL at T0 vs. 1307 ± 143.5 pg/mL at 12 M after KT). At 18 months post-KT, circulating sclerostin levels began to increase (1544 ± 287.8 pg/mL), being significantly higher at the moment of the second biopsy (24 months after KT) in comparison with the levels observed at 12 months (1307 ± 143.5 pg/mL at 12 M after KT vs. 2094 ± 318.6 pg/mL at 24 M after KT, *p* = 0.04).

Regarding circulating Dkk-1, 18 months after kidney transplant the levels of this bone biomarker suffered a significant reduction (1430 ± 283.1 pg/mL to 584.0 ± 168.2 pg/mL, *p* = 0.02) and remained reduced during the FW period until 24 months after transplantation (Figure 1B).

The circulating levels of free soluble RANK ligand (Figure 2A) were significantly increased at 12, 18 and 24 months (5.46 ± 1.09 *p* = 0.0002, 4.76 ± 0.82 *p* = 0.0004 and 4.33 ± 0.91 pg/mL *p* = 0.001, respectively) after kidney transplantation when compared to baseline levels (0.95 ± 0.29 pg/mL).

By contrast, the OPG circulating levels were significantly reduced at 12 months after KT (1192 ± 105.4 pg/mL at baseline vs. 819.2 ± 109.0 pg/mL, *p* = 0.04), being lower than baseline levels until the moment of the second biopsy (Figure 2B).

### 3.2. Evolution of Classic Bone Biomarkers after Kidney Transplantation

In addition to these data, serum levels of calcium (Figure 3A) were significantly higher at 12-, 18- and 24-months levels after transplant compared with baseline. By contrast, serum phosphorus levels (Figure 3B) were significantly lower after KT at 12, 18 months and 24 months, in comparison with baseline.

Intact PTH serum levels (Figure 3C) also decreased significantly during the post-transplant period, from 302.8 ± 57.6 at time of transplant to 115.9 ± 33.6 ng/L at 12 months, to 100.9 ± 11.5 ng7 L at 18 months (*p* = 0.02) and to 106.9 ± 14.5 ng/L at 24 months (*p* = 0.02) after transplant. Accordingly, FGF-23 circulating levels (Figure 3D) decreased from 190.5 ± 52.7 pg/mL at time of transplantation to 34.7 ± 13.4 pg/mL at 12 months, 32.3 ± 10.1 pg/mL at 18 months (*p* = 0.01) and 36.9 ± 12.3 pg/mL at 24 months after transplant.

### 3.3. Correlation between Novel Bone Biomarkers and Classic Bone Biomarkers

In univariate analysis, levels of circulating bioactive sclerostin did not correlate with any classic bone marker (Table 2). On the other hand, plasma levels of Dkk-1 correlated significantly and negatively with calcium and positively with intact PTH.

Plasma levels of free soluble RANK-ligand correlated significantly and positively with calcium and negatively with phosphorus and intact PTH. In addition, the levels of sRANK-L, negatively correlated with its decoy receptor, OPG (r = −0.64, *p* = 0.0002). By contrast, OPG plasma levels correlated significantly and negatively with calcium, and positively correlated with intact PTH and 25-OH-VitD without reaching statistical significance.

### 3.4. Correlation between Bone-Related Biomarkers and Bone Biopsy Parameters

Regarding bone parameters, it was found in univariate analysis (Table 3) that plasma bioactive sclerostin levels correlated significantly and positively with trabecular separation and negatively with trabecular number, both indicating that increased levels of this new biomarker are related to low bone volume. In other to assess the effects of other variables, a partial correlation was performed between sclerostin levels and bone biopsy parameters, controlling for the effect of classic bone biomarkers. The results expressed in Table 4 show that the correlation found between plasma bioactive sclerostin levels with trabecular separation and with trabecular number are in both cases independent from the levels of calcium, phosphorus, intact PTH and plasma creatinine.

### 3.5. Histologic Analysis

Figure 4 represents a baseline bone biopsy normal histology pattern with osteocytes (A) and the bone biopsy during the follow-up, with the evidence of osteocytes lacunes. The increase in osteocyte apoptosis after transplant is clear, observable by the existence of a large number of osteocytes lacunes in the bone biopsy.

## 4. Discussion

In this study, we aimed to prospectively evaluate the evolution of serum bone-related biomarkers after kidney transplantation, namely the emerging biomarkers: sclerostin, Dkk-1, RANKL and OPG. In addition, we intended to observe the correlation between these new biomarkers and classic bone biochemical parameters as well as bone biopsy parameters.

### 4.1. Evolution of Classic Bone Biomarkers after Kidney Transplantation

Patients with CKD or end-stage kidney disease (ESKD) suffer disturbances in calcium and phosphate metabolism, decreased calcitriol synthesis, increased synthesis and secretion of PTH and metabolic acidosis, which results in the presentation of histopathologic changes observed in bone, typically characterized by changes in bone turnover (can range from low to high), volume (can range from low to high), and mineralization (can be normal or abnormal) [7,12]. After KT, these biochemical abnormalities in PTH, calcium and phosphate resolve within the first year in most patients, and bone biopsy results show a reduction in bone activity, namely at cellular level, suggesting increased risk of adynamic bone and loss of bone volume [1,26,27].

Post renal transplantation, as our previous study [3] suggested, is characterized by the development of bone disease with low turnover and low bone volume, which is an important concern in this population. In our previous study [3] the biopsy performed after KT revealed a significant decrease in remodeling parameters compared with baseline biopsy, namely those related to bone formation, such as osteoblast surface/bone surface and osteoblast number/bone surface (data already published), accompanied by a marked reduction in bone formation rate [3]. The second biopsy in the follow-up also revealed a significant increase in trabecular separation and a significant decrease in trabecular number when compared with baseline bone biopsy. These results were accompanied by a slight reduction in bone volume that suggests loss in bone quantity [3].

Despite trans-iliac bone biopsy being considered the gold standard when assessing bone metabolism in CKD patients, new biomolecules have been recently associated with the development and progression of CKD-MBD.

### 4.2. Evolution of Serum Bone-Related Biomarkers after Kidney Transplantation

There has been recent discussion on two main systems that may explain the regulation of bone modelling/remodeling at the molecular level: (1) inhibitors of Wnt signaling pathway-mainly sclerostin but also secreted Dkk-1, and (2) RANK/RANK L/OPG system [28].

(1)Wnt signaling pathway: Sclerostin and Dkk-1

Our data showed a significant increase in plasma levels of bioactive sclerostin after KT accompanied by a significant reduction in plasma levels of Dkk-1, suggesting a promotion of the inhibition of bone formation by osteoblasts through the activation of these inhibitors of the Wnt signaling pathway. Before KT, circulating levels of bioactive sclerostin are high and immediately after KT these levels decrease, but after 12 months they start to increase, being significantly higher at 24 months after KT in comparison with the levels observed at 12 months. These results agree with Bonania et al. [29], which showed elevated serum sclerostin levels before KT and after 15 days of transplantation in all patients. With the improvement in renal function, sclerostin levels dropped, to increase again 6 and 12 months after engraftment. Interestingly, in another report, Araújo et al. demonstrated that, although sclerostin serum levels were decreased in the immediate post-KT period, levels measured in bone and SOST gene expression were actually increased [30]. Other studies have also shown that the serum concentration of sclerostin does not necessarily reflect sclerostin levels in bone [31,32]. The lower sclerostin levels immediately after transplantation can be explained by several hypotheses. First, sclerostin is a small, positively charged molecule, which in healthy people is filtered through the glomerular membrane but is mostly reabsorbed in the proximal tubule [33]. However, immediately after KT, tubular dysfunction can occur due to ischemia-reperfusion injury, which can induce overload proteinuria, thus causing the loss of high amounts of sclerostin in the urine [34]. Another important hypothesis is the glucocorticoid treatment used in KT, which appears to initially suppress serum sclerostin levels, but prolonged treatment appears to be associated with increased serum sclerostin levels [35]. In fact, there seems to be a disconnection between bone sclerostin and circulating sclerostin levels, caused by glucocorticoids [36,37]. Sustained increase in plasma levels of bioactive sclerostin after KT, occurs in parallel with the reduction of PTH, although we did not find a significant correlation between sclerostin and PTH, possibly due to small sample size. This inverse association between serum sclerostin and PTH concentrations is well described in the literature, with PTH being a regulator of sclerostin production, reducing its expression [13,34,38]. The increase in physical activity in these patients after KT may also help explain this decrease [38].

On the other hand, circulating Dkk-1 is elevated before KT, with a reduction in its levels occurring after KT. Although sclerostin and Dkk-1 are proteins related to decreased bone formation, by inhibiting Wnt signaling through their binding to LRP5/6 co-receptors, in our group of patients, while sclerostin increases, Dkk-1 decreases. This result can be explained by the mutual compensatory regulation that these molecules appear to have: when one of the molecules becomes more highly expressed, the other is selectively suppressed [39]. Although both act by binding to the same receptor, they bind at different sites on the receptor, have different mechanisms of action, pattern of expression and act on osteoblasts at different stages of development [12,35,39]. It has recently been shown that inhibition of Dkk-1 increases sclerostin expression, suggesting a potential compensatory mechanism that may be responsible for the weak anabolic effects of suppression of Dkk-1 [12]. We also observed a significant increase in the ratio of these two biomarkers, which reflects the balance between higher levels of sclerostin accompanied by lower levels of Dkk-1 at the moment of the second biopsy. This reinforces the view that these glycoproteins have different mechanisms and pattern of expression after KT.

The use of corticosteroids in post-KT may also explain the reduction in Dkk-1 levels, as has been previously shown [35].

Another possible explanation for increased sclerostin levels and decreased Dkk-1 is PTH, which appears to have an important role in regulating at the level of the SOST/Sclerostin gene [40]. In our study, plasma levels of Dkk-1 correlated significantly and negatively with calcium and positively with intact PTH. This correlation of Dkk-1 with PTH agrees with a study by Viapiana et al. [38], which demonstrated that unlike sclerostin, serum levels of Dkk-1 are significantly increased in primary hyperparathyroidism and significantly correlated with PTH levels. Different work [39,40] also demonstrated that, long-term treatment with teriparatide in postmenopausal women with osteoporosis, is associated with significant increases in serum Dkk-1 levels. This mechanism also explains the negative correlation between Dkk-1 and calcium, which does not appear to be a direct relationship, but rather a consequence of the correlation of the levels of Dkk-1 and PTH [27].

(2)RANK/RANK L/OPG system

In addition, the data also showed a significant increase in plasma levels of sRANKL after KT accompanied by a significant reduction in plasma levels of its decoy receptor OPG, suggesting an enhanced bone resorption by osteoclasts mediated by sRANKL. With the decrease in PTH, an increase in OPG and a decrease in RANKL might be expected, but the same results were observed in another study, where there was a decrease in serum OPG levels and an increase in RANKL after KT [30]. These results can have several explanations. First, overexpression of SOST in vivo leads to an increase in RANKL levels [41]. Other studies have already shown that SOST/sclerostin upwardly regulates RANKL, increases the number of osteoclasts and increases the RANKL/OPG ratio [30,42]. Taken together, these results suggest that the loss of bone volume observed in these patients after KT could be related to the inhibition of bone formation mediated by sclerostin accompanied by an enhanced bone resorption mediated by sRANKL. Another possibility for such a significant increase in RANKL in post-KT may be the use of corticosteroids during transplantation. The excess of corticosteroids promotes apoptosis of mature osteoblasts and osteocytes, [43,44,45] and recent studies show that osteocyte apoptosis increases RANKL expression in nearby healthy osteocytes [31,46,47]. On the other hand, our results also revealed that the increased plasma levels of sRANKL significantly correlated with the increase in serum calcium levels and with the reduction in serum phosphorus and intact PTH levels, suggesting that this new bone biomarker presents a good correlation with classic bone biomarkers in this group of transplanted patients.

### 4.3. Serum Bone-Related Biomarkers vs. Histomorphometric Analysis of Bone Biopsy

In our first study, we observed in the second biopsy, compared to baseline BB, a significant decrease in remodeling parameters, namely those related to bone formation, such as osteoblast surface/bone surface (Ob.S/BS) and osteoblast number/bone surface (N.Ob/BS). We also verified that the erosion surface/bone surface (ES/BS) and the mean osteoclast surface/bone surface (Oc.S/BS) was reduced in the FW biopsy, although without statistical significance. We also described a decrease in trabecular number (TbN) and an increase in trabecular separation (TbSp).

In this study, we found that the increased plasma bioactive sclerostin levels correlated with the increase in trabecular separation and with the reduction in trabecular number, both indicating that increased levels of this new biomarker is related with low bone volume by the inhibition of bone formation, independently of the levels of bone classic biomarkers. This study observed a negative correlation between bone volume and OPG, when a positive one would be expected. The mechanism of how OPG works in patients with CKD-MBD is not yet known. Studies show that, in this group of patients, high levels of OPG do not seem to protect the skeleton against increased bone resorption, and the role of OPG in bone metabolism does not seem to be consensual in this group of patients, even after transplantation [48,49]. More studies are needed to better understand the role of this biomolecule in these groups of patients.

We also found a significant negative correlation between sRANKL and the osteoclastic surface. Although we expect a positive correlation, some studies show that the process by which osteocytes support osteo-clastogenesis is not yet clear. RANKL appears to be able to activate osteoclasts through its soluble form (sRANKL) or by the membrane-bound form of RANKL. Results of experiments to clarify this process have shown that direct interactions between osteocytes and osteoclast precursors are necessary for efficient osteoclasto-genesis, with minimal sRANKL contribution [50,51]. This hypothesis seems to corroborate our results, because although we have remarkably high levels of sRANKL, the osteoclastic surface is reduced. Another finding in our study that supports this hypothesis is the clear increase in osteocyte apoptosis after KT, observable by the existence of a large number of osteocytes lacunes, and that can lead to a decrease in the membrane-bound form of RANKL. Further studies are needed to verify this hypothesis.

This study has some limitations such as the small number of patients and the lack of control group. Despite these, we can still draw relevant conclusions about the evolution of these novel biomarkers and their relationship with bone histomorphometric markers after KT.

## 5. Conclusions

These data obtained from Sclerostin, Dkk-1 and RANKL also seem to indicate that more importantly than an increased bone resorption, it seems that KT patients present a markedly suppressed bone formation. Taken together, these results suggest that the loss in bone volume observed after KT could be mainly related to the inhibition of bone formation mediated by sclerostin accompanied by an enhanced bone resorption mediated by sRANKL.

The present findings contribute to increasing evidence as to the role of these new bone biomarkers in clinical practice.

## 6. Highlights


Our study shows a significantly increase in the circulating levels of bioactive sclerostin after kidney transplant.Our prospective study reinforces the view that the loss in bone volume observed after kidney transplantation could be mainly related to the inhibition of bone formation mediated by sclerostin changes.Our data also reinforce the view that the enhanced bone resorption observed in the follow-up of kidney transplant appears to be mediated by the elevated circulating levels of sRANKL.


## Figures and Tables

**Figure 1 jcm-11-00457-f001:**
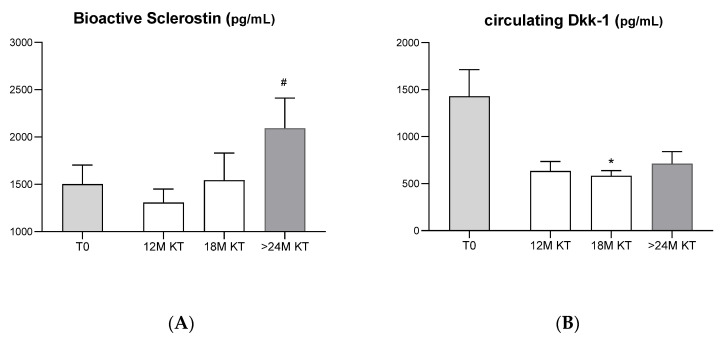
Circulating levels of bioactive Sclerostin (**A**) and Dicckopf-1 (**B**) at baseline (T0 before KT *n* = 13) and 12 (*n* = 6), 18 (*n* = 10) and 24 months (*n* = 6) after kidney transplantation. * *p* < 0.05, each time after TX (12, 18 and 24M) versus baseline (T0), and # *p* < 0.05, 24M after TX versus 12M after TX.

**Figure 2 jcm-11-00457-f002:**
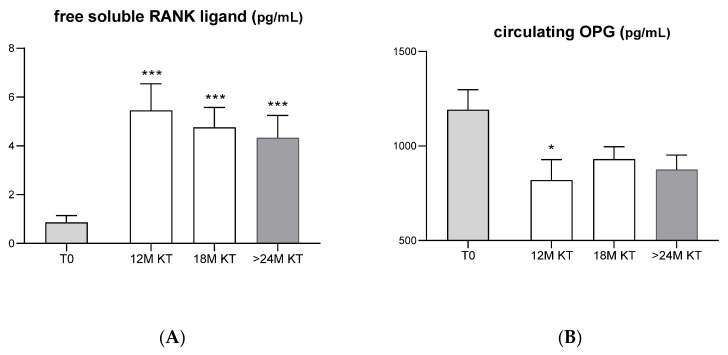
Circulating levels of free soluble RANK ligand (**A**) and osteo-protegerin (**B**) at baseline (T0 before KT *n* = 13) and 12 (*n* = 6), 18 (*n* = 10) and 24 months (*n* = 6) after kidney transplantation. * *p* < 0.05 and *** *p* < 0.001, each time after TX (12, 18 and 24M) versus baseline (T0).

**Figure 3 jcm-11-00457-f003:**
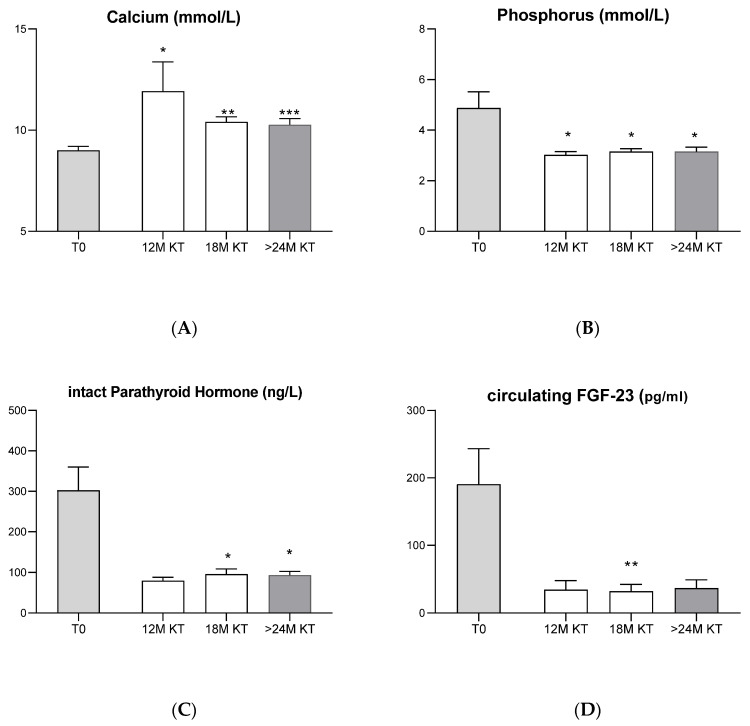
Serum levels of calcium (**A**), phosphorus (**B**), iPTH (**C**) and circulating levels of FGF-23 (**D**) at baseline (T0 before KT n = 13) and 12 (*n* = 6), 18 (*n* = 10) and 24 months (*n* = 6) after kidney transplantation. * *p* < 0.05, ** *p* < 0.01 and *** *p* < 0.001, each time after TX (12, 18 and 24M) versus baseline (T0).

**Figure 4 jcm-11-00457-f004:**
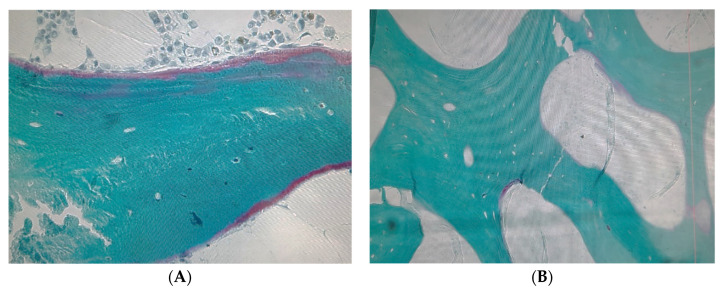
Bone biopsy. Baseline BB with Osteocytes (**A**) and FW BB with osteocytes lacunes (**B**). (Authors microscopy images 200×).

**Table 1 jcm-11-00457-t001:** Patient characteristics.

Patient	Sex	Age (Years)	Dialysis Mode	Primary Disease	Pcr (mg/dL) T0	Bone Related Pre-Transplant Medication	Immunosuppression	Pcr (mg/dL) T24	eGFR (mL/min) T24
**1**	**M**	**60**	**PD**	**HTN**	**1.83**	**Cinac**	**CsA + MMF + Pred**	**1.15**	**64**
**2**	**F**	**57**	**HD**	**HTN**	**1.54**	**Cinac + α-Calcid**	**CsA + MMF + Pred**	**0.91**	**48**
**3**	**M**	**59**	**HD**	**GN**	**1.74**	**-**	**CsA + MMF + Pred**	**1.5**	**48**
4	M	48	HD	GN	1.93	-	**CsA + MMF + Pred**	*	*
**5**	**M**	**34**	**HD**	**Unknown**	**2.45**	**Calcitriol**	**TAC + MMF + Pred**	**1.7**	**46**
**6**	**M**	**50**	**HD**	**IgAN**	**2.27**	**Cinac**	**CsA + MMF + Pred**	**1.9**	**37**
7	F	64	HD	Unknown	1.24	Calcitriol	CsA + MMF + Pred	*	*
8	F	63	HD	Unknown	1.68	Cinac	CsA + MMF + Pred	*	*
9	M	53	HD	Unknown	1.29	-	CsA + MMF + Pred	*	*
10	M	55	PH	Diab NP	2.18	Calcitriol	CsA + MMF + Pred	**	**
11	F	43	HD	GN	1.03	-	TAC + MMF + Pred	*	*
12	M	59	HD	Diab NP	1.98	α-Calcid	CsA + MMF + Pred	*	*
**13**	**M**	**33**	**HD**	**Unknown**	**2.03**	**-**	**TAC + MMF + Pred**	**1.24**	**65**

α-calcid: alfa-calcidol; cinac: cinacalcet; Diab NP: Diabetic Nephropathy; eGFR: estimated glomerular filtration rate; GN: Glomerulonephritis; HD: Hemodialysis; HTN: Hypertensive nephrosclerosis; IgAN: IgA Nephropathy; MMF: mycophenolate mofetil; PD: Peritoneal dialysis; Pcr: plasma creatinine; Pred: prednisone; TAC: tacrolimus. * Drop out to bone biopsy FW. ** No plasma. Patients in bold submitted to the 2nd bone biopsy.

**Table 2 jcm-11-00457-t002:** Correlation between novel bone biomarkers and classic bone biomarkers.

	Novel Bone Biomarkers			
Classic Bone Biomarkers	Sclerostin	Dkk-1	sRANK-L	OPG
Ca	r = −0.22; *p* = 0.25	r = −0.60; *p* = 0.0002 ***	r = 0.66; *p* < 0.0001 ****	r = −0.43; *p* = 0.02 *
Pi	r = −0.10; *p* = 0.75	r = 0.23; *p* = 0.1893	r = −0.44; *p* = 0.01 *	r = −0.10; *p* = 0.63
PTHi	r = −0.33; *p* = 0.15	r = 0.50; *p* = 0.01 *	r = −0.57; *p* = 0.005 **	r = 0.40; *p* = 0.07
25-OH-VitD	r = 0.52; *p* = 0.20	r = −0.59; *p* = 0.08	r = −0.00; *p* > 0.9999	r = −0.64; *p* = 0.07
FGF-23	r = 0.10; *p* = 0.73	r = 0.13; *p* = 0.46	r = −0.33; *p* = 0.07	r = −0.10; *p* = 0.58

* *p* < 0.05, ** *p* < 0.01, *** *p* < 0.001, **** *p* < 0.0001.

**Table 3 jcm-11-00457-t003:** Correlation between novel bone biomarkers and bone biopsy parameters.

	Novel Bone Biomarkers			
Bone Biopsy Parameters	Sclerostin	Dkk-1	sRANK-L	OPG
Bone volume (BV/TV)	r = −0.42; *p* = 0.18	r = −0.43; *p* = 0.11	r = 0.16; *p* = 0.58	r = −0.60; *p* = 0.02 *
Osteoblast surface (Ob.S/BS)	r = −0.35; *p* = 0.22	r = 0.003; *p* = 0.99	r = −0.50; *p* = 0.07	r = 0.10; *p* = 0.70
Osteoclast surface (Oc.S/BS)	r = 0.29; *p* = 0.32	r = −0.001; *p* = 0.99	r = −0.60; *p* = 0.02 *	r = 0.21; *p* = 0.44
Trabecular Separation (TbSp)	r = 0.76; *p* = 0.0055 **	r = 0.47; *p* = 0.08	r = −0.01; *p* = 0.97	r = 0.27; *p* = 0.34
Trabecular Number (TbN)	r = −0.78; *p* = 0.0043 **	r = −0.47; *p* = 0.08	r = 0.06; *p* = 0.83	r = −0.25; *p* = 0.39
Bone Formation Rate (BFR/BS)	r = 0.42; *p* = 0.27	r = 0.13; *p* = 0.70	r = −0.43; *p* = 0.21	r = −0.03; *p* = 0.94

* *p* < 0.05 and ** *p* < 0.01.

**Table 4 jcm-11-00457-t004:** Partial correlation between sclerostin levels and bone biopsy parameters, with controlling of the effect of other variables.

Spearman’s Correlations	Spearman’s Rho	*p*
**Sclerostin** **-** **trabecular separation (TbSp)**	0.762	0.006
Sclerostin-TbSp conditioned on variables: Ca	0.756	0.007
Sclerostin-TbSp conditioned on variables: Pi	0.795	0.003
Sclerostin-TbSp conditioned on variables: PTHi	0.703	0.035
Sclerostin-TbSp conditioned on variables: Pcreatinine	0.765	0.006
**Sclerostin-Trabecular Number (TbN)**	−0.776	0.005
Sclerostin-TbN conditioned on variables: Ca	−0.768	0.006
Sclerostin-TbN conditioned on variables: Pi	−0.807	0.003
Sclerostin-TbN conditioned on variables: PTHi	−0.742	0.022
Sclerostin-TbN conditioned on variables: Pcreatinine	−0.785	0.004

## Data Availability

The data presented in this study are available on request from the corresponding author.

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
