# Peer review of "Could Bone Biomarkers Predict Bone Turnover after Kidney Transplantation?—A Proof-of-Concept Study"

_jcm, 2022, doi:10.3390/jcm11020457_

Round 1

Reviewer 1 Report

Magalhaes and colleagues presented an interesting study, looking at the biomolecules association with CKD-MBD after kidney transplant. The authors were able to identify that the loss of bone volume after KT was mainly due to inhibition of bone formation mediated by sclerostin and by enhanced bone reabsorption mediated by sRANKL.

  • The patient enrollment was done in the year 2008? When the results were measured, recently or at the time of enrollment? How the samples were stored? Please include details in the methodology section.
  • There are many factors that affect the pre and post-transplant level of the biomarkers. It will be nice to see the baseline characteristics table for patients enrolled pre-KT. Such as diabetes status, other comorbidities, gender status, postmenopausal status in women, type of modality pre-transplant, etc.
  • How many patients were receiving bone-related medications pre and post-transplant? And how many developed bone disorders post-KT is not clear.
  • There are a few factors pre-KT that affect the level of post-KT biomarkers level and are important to consider when determining the level of bone turnover biomarkers. Such as level of these biomarkers pre-KT, gender, medications, modality differences, time after transplantation, etc. Do authors have a theory explaining the effects of any of these factors?
  • I would recommend running a linear mixed or other relevant models to identify the fixed effects of the aforementioned factors in relation to the higher biomarker levels post-KT.
  • In the results sclerostin did not correlate with any classic bone biomarkers, I would not suggest strongly mentioning that sclerostin mediated the inhibition of bone formation in the conclusion based on the results.
  • The sample size is extremely small, especially at follow-up, so I am not sure if there is enough power to run the significant analysis. The limitation paragraph should be included in the discussion section instead of the conclusion.
  • Figure 6 results should be moved to the results section from the discussion section. Also, the demographic results should be moved to the results section from the method section.
  • Do authors suggest any therapeutic strategies (pre and/or post-KT) for the prevention of bone disorder? The time of study seems a long time ago, many therapeutic and procedure strategies (pre and post-KT) have been improved since then, do authors think the results have different impacts based on the timeline? The details should be included in the discussion section.

Author Response

Magalhaes and colleagues presented an interesting study, looking at the biomolecules association with CKD-MBD after kidney transplant. The authors were able to identify that the loss of bone volume after KT was mainly due to inhibition of bone formation mediated by sclerostin and by enhanced bone reabsorption mediated by sRANKL.

  • The patient enrollment was done in the year 2008? When the results were measured, recently or at the time of enrollment? How the samples were stored? Please include details in the methodology section.

We enrolled thirteen consecutive patients submitted to deceased donor KT at our institution between 2008 and 2010. First biopsy was performed between 2008-2010 and the second bone biopsy was performed in 2010-2013. All samples of plasma were immediately stored at -80ºC, after collection. This information has been added to the text.  Over the years, the preservation of the samples with quality was guaranteed, since a preliminary test was carried out in which there was no protein degradation in relation to more recent samples.

  • There are many factors that affect the pre and post-transplant level of the biomarkers. It will be nice to see the baseline characteristics table for patients enrolled pre-KT. Such as diabetes status, other comorbidities, gender status, postmenopausal status in women, type of modality pre-transplant, etc.

Patient

Sex

Age (Years)

Dialysis Mode

Primary Disease

Bone Related Pre-Transplant Medication

Immunosuppression

1

M

60

PD

HTN

Cinac

CsA+MMF+Pred

2

F

57

HD

HTN

Cinac + α-Calcid

CsA+MMF+Pred

3

M

59

HD

GN

-

CsA+MMF+Pred

4

M

34

HD

Unknown

Calcitriol

TAC+MMF+Pred

5

M

53

HD

Unknown

-

CsA+MMF+Pred

6

M

48

HD

GN

-

CsA+MMF+Pred

 α-calcid: alfacalcidol; cinac: cinacalcet; GN: Glomerulonephritis; HD: Haemodialysis; HTN: Hypertensive nephrosclerosis; MMF: mycophenolate mofetil; PD: Peritoneal dialysis; Pred: prednisone; TAC: tacrolimus.

The authors added the requested data to the table. 

  • How many patients were receiving bone-related medications pre and post-transplant? And how many developed bone disorders post-KT is not clear.

Pre-transplant bone related medication is already described on the table added in the previous point. During follow-up, none of the patients was treated with calcium supplementation or active or native vitamin D.

We observed that all patients presented bone disorders after transplantation, with a decrease in bone activity, especially at the cellular level, which suggests that there is an increased risk of adynamic bone, as well as loss of bone volume.

  • There are a few factors pre-KT that affect the level of post-KT biomarkers level and are important to consider when determining the level of bone turnover biomarkers. Such as level of these biomarkers pre-KT, gender, medications, modality differences, time after transplantation, etc. Do authors have a theory explaining the effects of any of these factors?

The authors understand and thank the question raised by the reviewer. In this cohort we didn't observe differences between samples in pre-KT, as can be seen in the very low standard deviation values observed in biomarkers levels, revealing a high homogeneity between samples in the group.

  • I would recommend running a linear mixed or other relevant models to identify the fixed effects of the aforementioned factors in relation to the higher biomarker levels post-KT.

Having not identified any confounder factor we did not perform multivariate analysis.

  • In the results sclerostin did not correlate with any classic bone biomarkers, I would not suggest strongly mentioning that sclerostin mediated the inhibition of bone formation in the conclusion based on the results.

The authors understand and thank the question raised by the reviewer. There are several studies that demonstrate that classic bone biomarkers are poorly correlated with histomorphometry data from bone biopsies. In this sense, although we have not found a significant correlation with any classic bone biomarkers possibly due to the small sample size, we found that increased sclerostin levels were correlated with increased trabecular separation and reduced trabecular number, which indicates that this biomarker is related to low bone volume by inhibiting bone formation.

  • The sample size is extremely small, especially at follow-up, so I am not sure if there is enough power to run the significant analysis. The limitation paragraph should be included in the discussion section instead of the conclusion.

The authors understand that the weakness of this study is the small number of patients. Bone biopsy is an invasive procedure, causing minor complications such as pain and limited mobility in the first days, making it difficult to recruit more eligible patients for the study, especially at the time of the second biopsy, when we had a very high dropout rate. Although N is manifestly small, statistically significant results were found and the authors consider that they can still draw relevant conclusions about the evolution of these novel biomarkers and their relationship with bone histomorphometry markers after KT.  

The authors have included the limitation paragraph in the discussion section as suggested. 

  • Figure 6 results should be moved to the results section from the discussion section. Also, the demographic results should be moved to the results section from the method section.

The authors agreed with this recommendation and have moved figure 6 to the results section as suggested. The authors agreed with this recommendation and have moved demographic results to the results section as suggested.

  • Do authors suggest any therapeutic strategies (pre and/or post-KT) for the prevention of bone disorder? The time of study seems a long time ago, many therapeutic and procedure strategies (pre and post-KT) have been improved since then, do authors think the results have different impacts based on the timeline? The details should be included in the discussion section.

The authors consider that adequate control of secondary hyperparathyroidism in the dialysis phase, the prescription of calcium and vitamin D, as well as the use of a steroid-sparing regimen whenever possible can help prevent bone disorder. 

Regarding therapeutic and procedure strategies (pre and post-KT), the authors consider that the results would not be significantly different had the study taken place more recently. One of the major factors interfering with bone metabolism after KT is the use of corticosteroids. Although strategies aimed at minimizing or avoiding these agents are increasingly used, glucocorticoids are still administered to the majority of kidney transplant recipients worldwide. Moreover, the KDIGO 2017 guidelines did not significantly change the approach to the patient in the pre and post-KT. New anti-resorptive agents such as Denosumab have not been studied extensively in this population of kidney transplant recipients. The few existing studies appear to demonstrate that denosumab improves bone mineral density and bone quality in first-year KT. Our study seems to support it, as we found a significant increase in plasma levels of free sRANKL after KT accompanied by a significant reduction in plasma levels of its decoy receptor OPG, suggesting an enhanced bone reasorption by osteoclasts mediated by this mechanism. More studies are needed to better understand the impact of these new therapeutic strategies.

Reviewer 2 Report

Magalhaes and coworkers have submitted a very interesting article examining the relationship between novel bone biomarkers with traditional bone biomarkers and bone biopsy in patients who have had a kidney transplant.

It is well known that transplant recipients sustain bone loss and high fracture potential despite improvement in kidney function.  The mechanisms for the continued deterioration in bone health in this population are unknown; thus, studies such as these are needed.

It is admirable that the authors have been able to include individuals with bone biopsies, though as they admit, the numbers are relatively small.  The impact of this work could be strengthened by a number of considerations.

  1. The study population should be more fully characterized to include cause of kidney failure, length of time on dialysis, pre-transplant control of mineral metabolism, and whether or not any assessment of bone health was performed such as biopsy or DEXA.
  2. The authors refer to the many variables that may contribute to bone loss in this population.  it would be reasonable to parse these different issues more specifically in the results and discussion.  It is notable that these individuals were not on steroid sparing regimens.  What contribution do the authors think that 3400mg steroids over 6 months made to the bone pathology?  Likewise, most of the individuals who had two biopsies were on cyclosporin with only one on tacrolimus.  What contribution do these two calcineurin inhibitors make and would they have different effects on the Wnt signaling pathway?
  3. Please clarify if the studies were performed on all 13 patients who agreed to the study or only the 6 who underwent a second bone biopsy.
  4. With this small number of participants, presentation of the values for all participants would be reasonable, i.e., such as the authors present for Figure 5, where each subject's trabecular scores and sclerostin levels are indicated.  it would be useful to see the spread of values for the other parameters
  5. it is not clear from the text how bone biopsy results changed over time as the bone turnover parameters changed.
  6. Were DEXA scans or microCT scans performed?
  7. It does not appear that the authors performed the full standardized analysis of the bone biopsies (unless the reviewer missed it).  Did the subjects undergo tetracycline labeling?  Were the authors able to distinguish between changes in cancellous vs cortical bone?
  8. Do the authors have values for either total or bone specific alkaline phosphatase?

Author Response

Magalhaes and coworkers have submitted a very interesting article examining the relationship between novel bone biomarkers with traditional bone biomarkers and bone biopsy in patients who have had a kidney transplant.

It is well known that transplant recipients sustain bone loss and high fracture potential despite improvement in kidney function.  The mechanisms for the continued deterioration in bone health in this population are unknown; thus, studies such as these are needed.

It is admirable that the authors have been able to include individuals with bone biopsies, though as they admit, the numbers are relatively small.  The impact of this work could be strengthened by a number of considerations.

  1. The study population should be more fully characterized to include cause of kidney failure, length of time on dialysis, pre-transplant control of mineral metabolism, and whether or not any assessment of bone health was performed such as biopsy or DEXA.

Patient

Sex

Age (Years)

Dialysis Mode

Primary Disease

Bone Related Pre-Transplant Medication

Immunosuppression

1

M

60

PD

HTN

Cinac

CsA+MMF+Pred

2

F

57

HD

HTN

Cinac + α-Calcid

CsA+MMF+Pred

3

M

59

HD

GN

-

CsA+MMF+Pred

4

M

34

HD

Unknown

Calcitriol

TAC+MMF+Pred

5

M

53

HD

Unknown

-

CsA+MMF+Pred

6

M

48

HD

GN

-

CsA+MMF+Pred

α-calcid: alfacalcidol; cinac: cinacalcet; GN: Glomerulonephritis; HD: Haemodialysis; HTN: Hypertensive nephrosclerosis; MMF: mycophenolate mofetil; PD: Peritoneal dialysis; Pred: prednisone; TAC: tacrolimus.

 The authors have already added the requested data to the table in the revised version of the article. 

2. The authors refer to the many variables that may contribute to bone loss in this population.  it would be reasonable to parse these different issues more specifically in the results and discussion.  It is notable that these individuals were not on steroid sparing regimens.  What contribution do the authors think that 3400mg steroids over 6 months made to the bone pathology?  Likewise, most of the individuals who had two biopsies were on cyclosporin with only one on tacrolimus.  What contribution do these two calcineurin inhibitors make and would they have different effects on the Wnt signaling pathway?

It is known that one of the major factors interfering with bone metabolism after KT is the use of corticosteroids. In fact, steroids seem to have an early effect on osteoblast function and mineralization process, causing an inhibitory effect on bone formation and a delayed/impaired mineralization, but also being responsible for an increased osteoclastic activity. We believe that the results we obtained in the histomorphometric analysis of bone of KT recipients - reduction of bone activity and loss of bone volume - can be a consequence of transplant related medications, primarily corticosteroids. Calcineurin inhibitors have also been implicated, as they increase the risk of bone loss. Previous study showed that tacrolimus induced bone loss may be less severe in humans compared with that induced by CSA (evaluation by DXA), but these findings could be associated with the lower dose of glucocorticoids used in the tacrolimus group (Monegal A, et al. 2001 Calcif Tissue Int 68:83–86). We did not have enough sample size to compare the 2 groups.

3. Please clarify if the studies were performed on all 13 patients who agreed to the study or only the 6 who underwent a second bone biopsy.

13 patients were enrolled in baseline bone biopsy, however some were lost to the follow-up biopsy, and only 6 underwent the second biopsy. Bone biopsy is an invasive procedure, causing minor complications such as pain and limited mobility in the first days, making it difficult to recruit more eligible patients for the study, especially at the time of the second biopsy, when we had a very high dropout rate. Although N is manifestly small, statistically significant results were found and the authors consider that they can still draw relevant conclusions about the evolution of these novel biomarkers and their relationship with bone histomorphometric markers after KT. The authors have already included the limitation paragraph in the discussion section as suggested. 

4. With this small number of participants, presentation of the values for all participants would be reasonable, i.e., such as the authors present for Figure 5, where each subject's trabecular scores and sclerostin levels are indicated.  it would be useful to see the spread of values for the other parameters

After analyzing the suggestions of other reviewers, who highlighted that the findings in Figure 5 are already reported in Table 4, we chose to omit this Figure, as suggested.

5. it is not clear from the text how bone biopsy results changed over time as the bone turnover parameters changed.

The authors understand and thank the question raised by the reviewer. First biopsy was performed between 2008-2010 and second biopsy between 2010-2012. Initially, only the histomorphometry of bone biopsies was performed, which resulted in the first paper (Carvalho C, Magalhaes J, Pereira L, Simoes-Silva L, Castro-Ferreira I, Frazao JM. Evolution of bone disease after kidney transplantation: A prospective histomorphometric analysis of trabecular and cortical bone. Nephrology (Carlton). 2016;21(1):55-61), where we described a decrease in trabecular number (3.55 (1.81, 2.89) to 1.55/mm (1.24, 2.06), P = 0.018) and increase in trabecular separation (351.65 ± 135.04 to 541.79 ± 151.91 μm, P = 0.024) in follow-up biopsy, which suggest loss in bone quantity. This change represents the variation of bone quantity between the first biopsy (T0) and the second biopsy (24M). Small sample size can explain the lack of significance of the other data.

6. Were DEXA scans or microCT scans performed?

The authors understand and thank the question raised by the reviewer. Bone mineral density measurements (BMD), lumbar spine and femoral neck, were performed with DXA at the time of first bone biopsy and repeated with the second biopsy. The analysis of these data was discussed in more detail in our first paper (Carvalho C, Magalhaes J, Pereira L, Simoes-Silva L, Castro-Ferreira I, Frazao JM. Evolution of bone disease after kidney transplantation: A prospective histomorphometric analysis of trabecular and cortical bone. Nephrology (Carlton). 2016;21(1):55-61).

7. It does not appear that the authors performed the full standardized analysis of the bone biopsies (unless the reviewer missed it).  Did the subjects undergo tetracycline labeling?  Were the authors able to distinguish between changes in cancellous vs cortical bone?

As explained by the authors above, the analysis of these data was discussed in more detail in our first paper (Carvalho C, Magalhaes J, Pereira L, Simoes-Silva L, Castro-Ferreira I, Frazao JM. Evolution of bone disease after kidney transplantation: A prospective histomorphometric analysis of trabecular and cortical bone. Nephrology (Carlton). 2016;21(1):55-61). Opposite to our findings in trabecular bone, we did not find differences in cortical parameters before and after KT. Cortical formative parameters seem to correlate with trabecular correspondents. In our patients, these correlations were more consistent at the external cortex, including also a positive correlation with erosion parameters, which may suggest a stronger correlation between trabecular and external cortical remodelling process.

8. Do the authors have values for either total or bone specific alkaline phosphatase?

No, neither total or bone specific alkaline phosphatase were measured.

Reviewer 3 Report

Dear Authors,

Although it is well-written, my main concren is that it is very small adult cohort which weakens the study results.

There are several other concerns listed below:

What was the reason of too late reporting date of the patients (patients transplanted 13-14 years ago: 2008-2009

First evaluation for bone biomarkers and routine markers like Ca, P, PTH after KT is at 12th month. After the first year following KT, every 6 months these data were re-evaluated. As the first 12 months after KT is critical, particularly for P and PTH levels, these data should also be given.  

Why were not bone biopsy parameter change between the two histomorphometric measurements given in the manuscript in 6 patients?

Authors mentioned that serum and bone levels of these new bone biomarkers were not well correlated in previous studies. Why weren’t Sclerostin, Dkk1, RANKL and OPG staining performed to the bone bx specimens by using immunohistochemistry or by any other way?

Tables and figures:

Table 3.

Explanation for negative correlations between Oc surface and RANKL as well as bone volume and OPG is not satisfactory. These require further explanation.

Figure 3 has repetitive findings, so can be omitted.

Findings in Figure 5 are already reported in Table 4, so can be omitted.

Minor:

What is bone reabsorption…. meant bone resorption? Plase check in the whole text.

“Univariate analysis” term under correlation analysis subtitles should be corrected as correlation analysis. Univariate analysis belongs to regression analysis.

Author Response

Dear Authors,

Although it is well-written, my main concren is that it is very small adult cohort which weakens the study results.

There are several other concerns listed below:

What was the reason of too late reporting date of the patients (patients transplanted 13-14 years ago: 2008-2009

The authors understand and thank the question raised by the reviewer. Dr. Catarina Carvalho who started this work received her doctoral degree in 2017, and took a break from the research field, and in the last two years in the scope of Master thesis, Dr Juliana Magalhães decided to further explore this investigation.

The first biopsy was performed between 2008-2010 and the second bone biopsy was performed in 2010-2013. All samples of plasma were immediately stored at -80ºC, after collection. Over the years, the preservation of the samples with quality was guaranteed, since a preliminary test was carried out in which there was no protein degradation in relation to more recent samples.

First evaluation for bone biomarkers and routine markers like Ca, P, PTH after KT is at 12th month. After the first year following KT, every 6 months these data were re-evaluated. As the first 12 months after KT is critical, particularly for P and PTH levels, these data should also be given.  

T0 Transplant

1st Biopsy (~2mo)

2st Biopsy

Patient

Ca

(mmol/L)

Pi

(mmol/L)

PTH

(ngl/L)

Ca

(mmol/L)

Pi

(mmol/L)

PTH

(ngl/L)

Ca

(mmol/L)

Pi

(mmol/L)

PTH

(ngl/L)

1

2.40

2.42

495.7

2.65

0.67

119.8

2.75

1.00

117.8

2

2.50

1.18

757.1

2.70

0.46

119.5

2.85

0.97

93.9

3

2.10

0.95

108.5

2.45

0.43

86.3

2.50

0.90

71.8

4

2.10

2.48

55.3

2.65

0.60

22.6l

2.45

1.00

69.3

5

2.00

1.71

325.3

2.50

3.28

86.2

2.50

1.20

63.4

6

2.40

0.98

264.5

2.50

1.06

160.8

2.60

1.13

141.3

The authors understand and thank the question raised by the reviewer. The authors analysed data referring to 6 months after KT, but no major differences were found between these data and data from 12 months after KT. The analysis of these data was discussed in more detail in our first paper (Carvalho C, Magalhaes J, Pereira L, Simoes-Silva L, Castro-Ferreira I, Frazao JM. Evolution of bone disease after kidney transplantation: A prospective histomorphometry analysis of trabecular and cortical bone. Nephrology (Carlton). 2016;21(1):55-61). 

Why were not bone biopsy parameter change between the two histomorphometric measurements given in the manuscript in 6 patients?

The authors understand and thank the question raised by the reviewer. First biopsy was performed between 2008-2010 and second biopsy between 2010-2012. Initially, only the histomorphometry of bone biopsies was performed, which resulted in the first paper (Carvalho C, Magalhães J, Pereira L, Simões-Silva L, Castro-Ferreira I, Frazão JM. Evolution of bone disease after kidney transplantation: A prospective histomorphometric analysis of trabecular and cortical bone. Nephrology (Carlton). 2016;21(1):55-61), where we described a decrease in trabecular number (3.55 (1.81, 2.89) to 1.55/mm (1.24, 2.06), P = 0.018) and increase in trabecular separation (351.65 ± 135.04 to 541.79 ± 151.91 μm, P = 0.024) in follow-up biopsy, which suggest loss in bone quantity. This change represents the variation of bone quantity between the first biopsy (T0) and the second biopsy (24M). Small sample size can explain the lack of significance of the other data.

Authors mentioned that serum and bone levels of these new bone biomarkers were not well correlated in previous studies. Why weren’t Sclerostin, Dkk1, RANKL and OPG staining performed to the bone bx specimens by using immunohistochemistry or by any other way?

The authors understand and thank the question raised by the reviewer. Initially, it was our intention to use immunohistochemistry methods in bone biopsy samples, however in laboratory practice, this was not possible, since the type of methylmethacrylate used in the processing of bone biopsies is not easily removed in order to apply the technique of immunohistochemistry. We are still trying to optimize this technique.

Tables and figures:

Table 3.

Explanation for negative correlations between Oc surface and RANKL as well as bone volume and OPG is not satisfactory. These require further explanation.

The authors understand the question raised by the reviewer. As noted in the discussion, the process by which osteocytes support osteoclastogenesis is not yet clear. Although RANKL exists in its soluble form (sRANKL), the majority of RANKL is cell bound, and its membrane-bound form seems to play a more important role for osteoclast activation than its soluble form. This theory seems to be supported by images of bone biopsies before and after transplantation, where there is a clear increase in osteocyte apoptosis after KT, observable by the existence of a large number of lacunes osteocytes, which may indicate a decrease in the membrane-bound form of RANK. In this sense although we have remarkably high levels of sRANKL, the osteoclastic surface is reduced. The authors understand that further studies are needed to study this hypothesis.

The mechanism of how OPG acts in patients with CKD-MBD is not yet known, as what several studies demonstrate is that in this group of patient’s high levels of OPG do not seem to protect the skeleton against increased bone resorption. In the literature, the role of OPG in bone metabolism seems to be non-consensual in this group of patients, even after transplantation, appearing to play a more important role in vascular calcification. Another study was able to demonstrate that OPG also exists in two forms: it might exist either simply as a membrane-bound form or as an authentic transmembrane form. These results seem to suggest that OPG, which is in osteoclast precursors, may have a new role in osteoclast differentiation, in addition to its role as a soluble decoy receptor. More studies are needed to better understand the role of this biomolecule.

-Coen G, Ballanti P, Balducci A, Calabria S, Fischer MS, Jankovic L, Manni M, Morosetti M, Moscaritolo E, Sardella D, Bonucci E 2002 Serum osteoprotegerin and renal osteodystrophy. Nephrol Dial Transplant 17:233–238

-Rogers A, Eastell R. Circulating osteoprotegerin and receptor activator for nuclear factor kappaB ligand: clinical utility in metabolic bone disease assessment. J Clin Endocrinol Metab. 2005 Nov;90(11):6323-31. 

-Ralf Westenfeld, Markus Ketteler, Vincent M. Brandenburg, Anti-RANKL therapy—implications for the bone-vascular-axis in CKD? Denosumab in post-menopausal women with low bone mineral density *Comment on McClung MR, Lewiecki EM, Cohen SB et al. Denosumab in postmenopausal women with low bone mineral density. N Engl J Med 2006; 354: 821–831, Nephrology Dialysis Transplantation, Volume 21, Issue 8, August 2006, Pages 2075–2077.

-Woo, K., Choi, Y., Ko, SH. et al. Osteoprotegerin is present on the membrane of osteoclasts isolated from mouse long bones. Exp Mol Med 34, 347–352 (2002). 

Figure 3 has repetitive findings, so can be omitted.

The authors agreed with this recommendation and omitted Figure 3 as suggested.

Findings in Figure 5 are already reported in Table 4, so can be omitted.

The authors agreed with this recommendation and omitted Figure 5 as suggested.

Minor:

What is bone reabsorption…. meant bone resorption? Plase check in the whole text.

 The authors agreed with this recommendation and corrected as suggested. 

“Univariate analysis” term under correlation analysis subtitles should be corrected as correlation analysis. Univariate analysis belongs to regression analysis.

The authors agreed with this recommendation and corrected as suggested.

Round 2

Reviewer 2 Report

The reviewer thanks the authors for their responses.  In particular, it was not clear to this reviewer that the current manuscript was considered a follow up to a previous study.  The authors had alluded to that study in their introduction but the reviewer recommends that the authors are much more explicit in explaining that the authors are reviewing previously published histochemistry data to compare with newly measured serum biomarkers. 

Additionally, it is clearer to this reviewer that there are actually 2 different goals for this publication, the first being to compare the histochemistry data with newer serum biomarkers and second to compare the longitudinal evolution of new and established biomarkers.  Again, the authors should state these goals explicitly.

  1. The reviewer went back to the original publication and noted some discrepancies between the original report and the current report.  In the first report, the authors describe 7 individuals who underwent a second biopsy, while they refer to 6 in the current manuscript.
  2. The authors state that the kidney function was OK but they should include creatinine and eGFR over time for these individuals.  The authors indicate that they had measured these in the methods section but this reviewer did not see the values presented.
  3. The small number of subjects is acknowledged by the authors.  They differ in age, gender, underlying disease, dialysis modality, and pre-transplant medications.  This reviewer believes that presentation of the individual data included with the bar graph is important to show.  The reviewer has no doubt that the results will still be significant; however, demonstration of the diversity of response is also important information in this setting.  As the authors are aware, CKD-MBD is an extraordinarily complex and diverse entity.  In the current form, it is not possible for the reviewer to assess whether the data are distributed normally or not.
  4. The bone histology seems quite abbreviated. Admittedly, much of these data were presented and discussed in the first manuscript.  For this manuscript, it would be reasonable to at least review what had been previously reported.  The table does not indicate whether the bone biopsy parameters compared to the novel biomarkers represent the first biopsy, the second biopsy or some analysis of the difference in the biopsies.  this should be clarified.
  5. Did the authors stain the bone biopsies for sclerostin, a step that could address one of their discussion points?

Author Response

We sincerely appreciate all comments and suggestions raised by the 3 reviewers. We would like to emphasize the rigor and seriousness in the way we tried to answer all the questions.

This review process has had confused contours, as on 16th we received the major revision with comments of two reviewers (reviewer 1 and 3), and on the 20th, we submitted the revised version of the manuscript with the point-to-point response to these 2 reviewers. The manuscript was changed according to the comments of these 2 reviewers.

After that, on the 21st we received a 3rd report that was the revision of reviewer 2, who made valuable comments and to whom we answered in the following day. The revised version finally submitted on that day took into consideration the previous comments of all 3 reviewers. 

We sincerely hope that with our response to Round 2, all doubts will be clarified.

The reviewer thanks the authors for their responses.  In particular, it was not clear to this reviewer that the current manuscript was considered a follow up to a previous study.  The authors had alluded to that study in their introduction but the reviewer recommends that the authors are much more explicit in explaining that the authors are reviewing previously published histochemistry data to compare with newly measured serum biomarkers. 

Additionally, it is clearer to this reviewer that there are actually 2 different goals for this publication, the first being to compare the histochemistry data with newer serum biomarkers and second to compare the longitudinal evolution of new and established biomarkers.  Again, the authors should state these goals explicitly.

“This study aimed to prospectively evaluate and correlate the results of histomorphometric analysis of bone biopsies after kidney transplantation, observed and discussed in our first paper, with emerging serum biomarkers of the CKD-MBD spectrum: Sclerostin, Dkk-1, sRANKL and OPG. Also was our aim to compare the longitudinal evolution of these new biomolecules with established biomarkers.”

This paragraph was added in the final of Introduction.

1. The reviewer went back to the original publication and noted some discrepancies between the original report and the current report.  In the first report, the authors describe 7 individuals who underwent a second biopsy, while they refer to 6 in the current manuscript.

The authors understand the question raised by the reviewer. Although in the first paper we have 7 patients in FW, in this one we only have 6 since one of the patients did not have enough plasma samples to be included in this study, unfortunately.

2. The authors state that the kidney function was OK but they should include creatinine and eGFR over time for these individuals.  The authors indicate that they had measured these in the methods section but this reviewer did not see the values presented.

The authors added the Pcr at the T0 and T24, and eGFR at T24 to the table. 

Unfortunately, the authors recognize that in the elaboration of table 1 there was a transcription error, and therefore there were discrepancies between the first table of the first paper and this table.

 Corrections were made. 

Table 1: Patient’s characteristics.

Patient

Sex

Age (Years)

Dialysis

Mode

Primary

Disease

Pcr

(mg/dL)

T0

Bone Related

Pre-Transplant Medication

Immunosuppression

Pcr

(mg/dL)

T24

eGFR

(mL/min)

T24

1

M

60

PD

HTN

1.83

Cinac

CsA+MMF+Pred

1.15

64

2

F

57

HD

HTN

1.54

Cinac + α-Calcid

CsA+MMF+Pred

0.91

48

3

M

59

HD

GN

1.74

-

CsA+MMF+Pred

1.5

48

4

M

48

HD

GN

1.93

-

CsA+MMF+Pred

*

*

5

M

34

HD

Unknown

2.45

Calcitriol

TAC+MMF+Pred

1.7

46

6

M

50

HD

IgAN

2.27

Cinac

CsA+MMF+Pred

1.9

37

7

F

64

HD

Unknown

1.24

Calcitriol

CsA+MMF+Pred

*

*

8

F

63

HD

Unknown

1.68

Cinac

CsA+MMF+Pred

*

*

9

M

53

HD

Unknown

1.29

-

CsA+MMF+Pred

*

*

10

M

55

PH

Diab NP

2.18

Calcitriol

CsA+MMF+Pred

**

**

11

F

43

HD

GN

1.03

-

TAC+MMF+Pred

*

*

12

M

59

HD

Diab NP

1.98

α-Calcid

CsA+MMF+Pred

*

*

13

M

33

HD

Unknown

2.03

-

TAC+MMF+Pred

1.24

65

α-calcid: alfacalcidol; cinac: cinacalcet; Diab NP: Diabetic Nephropathy; eGFR: estimated glomerular filtration rate; GN: Glomerulonephritis; HD: Haemodialysis; HTN: Hypertensive nephrosclerosis; IgAN: IgA Nephropathy; MMF: mycophenolate mofetil; PD: Peritoneal dialysis; Pcr: plasma creatinine; Pred: prednisone; TAC: tacrolimus. *Drop out to bone biopsy FW. ** No plasma. Patients in bold submitted to the 2nd bone biopsy.

3. The small number of subjects is acknowledged by the authors.  They differ in age, gender, underlying disease, dialysis modality, and pre-transplant medications.  This reviewer believes that presentation of the individual data included with the bar graph is important to show.  The reviewer has no doubt that the results will still be significant; however, demonstration of the diversity of response is also important information in this setting.  As the authors are aware, CKD-MBD is an extraordinarily complex and diverse entity.  In the current form, it is not possible for the reviewer to assess whether the data are distributed normally or not.

The authors understand and thank the question raised by the reviewer. In figures 1 and 2 are represented circulating levels of bone biomarkers in bar graphs as mean levels in each time and as can be seen in the very low standard deviation values observed,the results reveal a high homogeneity between samples in each group. Despite patients enrolled in the study differ in age, gender, underlying disease, dialysis modality and pre-transplant medications, any of these factors was found as confounders in this cohort, and none change with discrepancy the levels of these biomarkers in each time of follow-up.

4. The bone histology seems quite abbreviated. Admittedly, much of these data were presented and discussed in the first manuscript.  For this manuscript, it would be reasonable to at least review what had been previously reported.  The table does not indicate whether the bone biopsy parameters compared to the novel biomarkers represent the first biopsy, the second biopsy or some analysis of the difference in the biopsies.  this should be clarified.

The authors agreed with this recommendation and a paragraph was added to the text with this data, as suggested by the reviewer.

“In our first study, we observed in the second biopsy, comparing to baseline BB, a significant decrease in remodeling parameters, namely those related to bone formation, such as osteoblast surface/bone surface (Ob.S/BS) and osteoblast number/bone surface (N.Ob/BS). We also verified that the erosion surface/bone surface (ES/BS) was reduced and the mean osteoclast surface/bone surface (Oc.S/BS) was reduced in the FW biopsy, although without statistical significance. We also described a decrease in trabecular number (TbN) and increase in trabecular separation (TbSp).

Table 3 represents the correlation between novel bone biomarkers and bone biopsy parameters, which include the results overtime, at baseline bone biopsy and at 24M, time of second biopsy.

5. Did the authors stain the bone biopsies for sclerostin, a step that could address one of their discussion points?

The authors understand and thank the question raised by the reviewer. Initially, it was our intention to use immunohistochemistry methods in bone biopsy samples, however in laboratory practice, this was not possible, since the type of methylmethacrylate used in the processing of bone biopsies is not easily removed in order to apply the technique of immunohistochemistry. We are still trying to optimize this technique. 

Reviewer 3 Report

Thank you.

Your explanations are not concordant with your revision in the paper. Again, there are several discrepancies between methods parts of your two studies produced from the same group of patients (in the 2016 paper and in this paper), particularly on the use of cinacalcet and anti-resorptive drugs. This is an important problem in my point of view.

Author Response

Thank you.

Your explanations are not concordant with your revision in the paper. Again, there are several discrepancies between methods parts of your two studies produced from the same group of patients (in the 2016 paper and in this paper), particularly on the use of cinacalcet and anti-resorptive drugs. This is an important problem in my point of view.

Authors‘ response

We sincerely appreciate all comments and suggestions raised by the 3 reviewers. We would like to emphasize the rigor and seriousness in the way we tried to answer all the questions.

This review process has had confused contours, as on 16th we received the major revision with comments of two reviewers (reviewer 1 and 3), and on the 20th, we submitted the revised version of the manuscript with the point-to-point response to these 2 reviewers. The manuscript was changed according to the comments of these 2 reviewers.

After that, on the 21st we received a 3rd report that was the revision of reviewer 2, who made valuable comments and to whom we answered in the following day. The revised version finally submitted on that day took into consideration the previous comments of all 3 reviewers. 

We sincerely hope that with our response to Round 2, all doubts will be clarified.

The authors understand that the methods need to be clearer, so some data has been added and others revised.

Regarding the use of cinacalcet and anti-resorptive drugs, the following sentence was added to methods section:

”Of these, due to severe hypercalcemia (3 mmol/L), one of the patients started cinacalcet (30 mg/day) at month 20. One patient took bisphosphonates for a short period of 6 months because of low mineral density on ray absorptiometry -Double X (DXA).”

Unfortunately, the authors recognize that in the elaboration of table 1 there was a transcription error, and therefore there were discrepancies between the first table of the first paper and this table.  Corrections were made. 

The authors take the opportunity to clarify that although in the first paper we have 7 patients in FW, in this one we only have 6 since one of the patients did not have enough plasma samples to be included in this study, unfortunately.

Also, according to a comment of the reviewer, the following sentence was added to discussion section. 

“In this study, observed a negative correlation between bone volume and OPG, when a positive one would be expected. The mechanism of how OPG works in patients with CKD-MBD is not yet known. Studies show that, in this group of patients, high levels of OPG do not seem to protect the skeleton against increased bone resorption, and the role of OPG in bone metabolism does not seem to be consensual in this group of patients, even after transplantation. More studies are needed to better understand the role of this biomolecule in these groups of patients.”

Table 1: Patient’s characteristics.

Patient

Sex

Age (Years)

Dialysis

Mode

Primary

Disease

Pcr

(mg/dL)

T0

Bone Related

Pre-Transplant Medication

Immunosuppression

Pcr

(mg/dL)

T24

eGFR

(mL/min)

T24

1

M

60

PD

HTN

1.83

Cinac

CsA+MMF+Pred

1.15

64

2

F

57

HD

HTN

1.54

Cinac + α-Calcid

CsA+MMF+Pred

0.91

48

3

M

59

HD

GN

1.74

-

CsA+MMF+Pred

1.5

48

4

M

48

HD

GN

1.93

-

CsA+MMF+Pred

*

*

5

M

34

HD

Unknown

2.45

Calcitriol

TAC+MMF+Pred

1.7

46

6

M

50

HD

IgAN

2.27

Cinac

CsA+MMF+Pred

1.9

37

7

F

64

HD

Unknown

1.24

Calcitriol

CsA+MMF+Pred

*

*

8

F

63

HD

Unknown

1.68

Cinac

CsA+MMF+Pred

*

*

9

M

53

HD

Unknown

1.29

-

CsA+MMF+Pred

*

*

10

M

55

PH

Diab NP

2.18

Calcitriol

CsA+MMF+Pred

**

**

11

F

43

HD

GN

1.03

-

TAC+MMF+Pred

*

*

12

M

59

HD

Diab NP

1.98

α-Calcid

CsA+MMF+Pred

*

*

13

M

33

HD

Unknown

2.03

-

TAC+MMF+Pred

1.24

65

α-calcid: alfacalcidol; cinac: cinacalcet; Diab NP: Diabetic Nephropathy; eGFR: estimated glomerular filtration rate; GN: Glomerulonephritis; HD: Haemodialysis; HTN: Hypertensive nephrosclerosis; IgAN: IgA Nephropathy; MMF: mycophenolate mofetil; PD: Peritoneal dialysis; Pcr: plasma creatinine; Pred: prednisone; TAC: tacrolimus. *Drop out to bone biopsy FW. ** No plasma. Patients in bold submitted to the 2nd bone biopsy.
